# Epithelial-Immune Cell Crosstalk in Salivary Gland Tumors: Implications for Tumor Progression and Diagnostic Assessment

**DOI:** 10.3390/ijms262010199

**Published:** 2025-10-20

**Authors:** Martina Sausa, Giuseppe Vergilio, Rosario Barone, Rossana Porcasi, Prince Ofori, Fatima Azhraa Haddad, Francesca Rappa, Francesca Levi-Schaffer, Angelo Leone

**Affiliations:** 1Department of Biomedicine, Neurosciences and Advanced Diagnostics, University of Palermo, 90127 Palermo, Italy; martina.sausa@gmail.com (M.S.); giuseppe.vergilio@unipa.it (G.V.); rosario.barone@unipa.it (R.B.); 2Department of Health Promotion, Mother and Child Care, Internal Medicine and Medical Specialties (PROMISE), University of Palermo, 90127 Palermo, Italy; r.porcasi@libero.it; 3Pharmacology and Experimental Therapeutics Unit, Institute for Drug Research, School of Pharmacy, Faculty of Medicine, The Hebrew University of Jerusalem, Jerusalem 91120, Israel; prince.ofori@mail.huji.ac.il (P.O.); francescal@ekmd.huji.ac.il (F.L.-S.); 4Department of Otolaryngology Head and Neck Surgery, Hadassah Medical Center, Hebrew University of Jerusalem, Jerusalem 91120, Israel

**Keywords:** salivary glands, pleomorphic adenoma, squamous cells carcinoma, mast cell tryptase, CD300a, mast cells

## Abstract

This study explores immunophenotypic and angiogenic profiles in salivary gland tumors (SGTs), focusing on epithelial–mesenchymal dynamics and immune–stromal interactions. Immunohistochemical analysis of E-cadherin, Vimentin, mast cell tryptase (MCT), CD300a, CK18, CD31, and vascular endothelial growth factor (VEGF) was performed in normal salivary tissue, pleomorphic adenomas (PA), and squamous cell carcinomas (SCCs) to assess epithelial plasticity, mast cell (MC) involvement, and vascular remodeling. Normal glands showed compartmentalized E-cadherin (epithelial) and Vimentin (mesenchymal) expression, with stromal MCs positive for MCT and CD300a. PA exhibited reduced E-cadherin, increased Vimentin, and atypical co-localization of CK18 with MCT/CD300a in ductal cells, indicating immune–epithelial plasticity. SCC displayed epithelial–mesenchymal transition (EMT), architectural disruption, and reduced MCT/CD300a. Notably, diminished MCT may reflect either decreased MCs density or prior degranulation, with possible diffuse MCT in stroma. Angiogenic profiling showed maximal CD31 in PA and minimal in SCC, while VEGF peaked in normal tissue, suggesting deregulated angiogenesis. SGT progression involves immune–epithelial plasticity, vascular deregulation, and stromal reprogramming. Immune marker localization within epithelial cells challenges histogenetic models and may inform prognostic assessment and targeted therapeutic strategies.

## 1. Introduction

Salivary gland tumors (SGTs) arise from the epithelial cells of major (parotid, submandibular, sublingual) and minor glands, displaying remarkable histological heterogeneity that reflects the intrinsic plasticity of these secretory structures. The global incidence of salivary gland tumors (SGTs) is relatively low, estimated between 0.4 and 13.5 cases per 100,000 individuals per year. Benign tumors predominate; however, malignant salivary gland cancers (SGCs) account for approximately 0.4 to 2.6 cases per 100,000. The incidence shows geographical variability and tends to be higher in Western countries, where rates may reach 2.5–3.0 per 100,000. Benign forms, such as pleomorphic adenoma (PA), the most common SGT, show a biphasic proliferation of epithelial and mesenchymal elements. At the opposite extreme lies squamous cell carcinoma (SCC), a rare but aggressive variant, characterized by atypical squamous differentiation and a markedly infiltrative phenotype [1,2]. SCCs may develop as primary neoplasms, uncommon in all regions, or more frequently as secondary involvement from cutaneous SCCs of the head and face, whose epidemiology reflects chronic ultraviolet exposure. Regardless of origin, these tumors are locally invasive, metastasize early, and respond poorly to radiotherapy and chemotherapy [3]. Although SGTs account for only 2–4% of head and neck tumors, the incidence of SCC, particularly secondary forms, is increasing worldwide, with poor prognosis mainly due to late diagnosis, lack of predictive biomarkers, and limited therapeutic options [4]. This scenario underscores the need to clarify molecular mechanisms of progression to identify novel diagnostic and therapeutic targets. A central process in SGT carcinogenesis is epithelial–mesenchymal transition (EMT), through which epithelial cells lose polarity and adhesion, acquiring migratory, apoptosis-resistant traits. EMT is orchestrated by conserved signaling pathways (e.g., TGF-β, Wnt/β-catenin, Notch) and transcription factors (Snail, Slug, ZEB1/2) that repress E-cadherin, a key adhesion molecule. E-cadherin downregulation, coupled with Vimentin upregulation, the so-called “cadherin switch”, is a hallmark of EMT and correlates with invasiveness and metastatic potential [5,6,7]. However, the contribution of the tumor microenvironment (TME) to EMT modulation in SGTs remains poorly understood. The salivary gland TME includes cancer-associated fibroblasts (CAFs), immune cells, extracellular matrix (ECM), and soluble mediators [8]. Among immune cells, mast cells (MCs) are multifunctional players that, beyond their classical role in allergy, release mediators such as tryptase (MCT), histamine, VEGF, and TNF-α, thereby influencing angiogenesis, ECM remodeling, and EMT [9,10]. MCT, a serine protease stored in MC granules, activates protease-activated receptors (PARs) on epithelial cells, promoting invasive signaling cascades [11,12]. In SGTs, MC infiltration correlates with E-cadherin loss and invasive growth [9,13]. MC activity is modulated by CD300a, an inhibitory receptor expressed on myeloid and lymphoid cells that binds phosphatidylserine and phosphatidylethanolamine to suppress degranulation and cytokine release [14,15,16]. Its role in cancer is context-dependent: in some settings (e.g., hepatocellular carcinoma) it favors immunosuppression and metastasis, whereas in others (e.g., non-small cell lung cancer) it limits tumor progression [17,18,19]. Notably, aberrant CD300a expression has also been reported in epithelial cells of PA and SCC, suggesting a role in immune–epithelial crosstalk [20].

Tumor-associated macrophages (TAMs), particularly the M2 subtype, also contribute to immune evasion and progression via secretion of anti-inflammatory cytokines and promotion of angiogenesis [21,22]. M2 macrophages express CD300a, further enhancing their immunosuppressive functions and creating a permissive TME.

This study investigates the histological and immunohistochemical profiles of MCT and CD300a in healthy salivary tissues, PA, and SCC, with emphasis on their interplay in EMT and TME regulation. We hypothesize that: (1) MC infiltration and MCT release correlate with the loss of E-cadherin and acquisition of a mesenchymal phenotype; (2) CD300a modulates the activity of MC and epithelial cells, acting as a context-dependent immune checkpoint (inhibitory in PA, protumorigenic in SCC); and (3) Co-expression of MCT and CD300a in ductal cells predicts aggressiveness and metastatic potential. The integrated analysis of MCT and CD300a may reveal novel diagnostic and therapeutic opportunities. Strategies such as tryptase inhibition (e.g., nafamostat) or CD300a modulation could counteract EMT and restore immune–epithelial homeostasis. Identifying tumor subpopulations with aberrant immune marker expression may also guide personalized immunotherapies tailored to salivary gland TME heterogeneity [23,24].

## 2. Results

### 2.1. Evolutionary Dynamics of Salivary Neoplasms: Structural Alterations and Cell Plasticity from Glandular Physiology to Malignant Transformation

Histological analysis of healthy submandibular glands revealed a highly organized glandular architecture, with a clear demarcation between epithelial and mesenchymal compartments (Figure 1A–C). The serous and mucous acini, identified by hematoxylin-eosin (H&E) staining, showed a regular lobular arrangement, bordered by connective stroma rich in fibroblasts and blood vessels (Figure 1A–C). The PA presented a distinctive histological heterogeneity, with biphasic proliferation of epithelial cells and mesenchymal stroma (Figure 1D–F). The epithelial components included tubular ductal structures, cystic and areas of squamous metaplasia, while the stroma showed myxoid, chondroid and hyaline differentiation (Figure 1D–F). Finally, in SCC, the glandular architecture was completely absent, with replacement by infiltrating nests of atypical squamous cells and desmoplastic and inflammatory stroma (Figure 1G–I).

### 2.2. Progressive Alteration in the Expression of Epithelial and Mesenchymal Markers

To understand whether neoplastic progression was associated with a progressive alteration in the expression of epithelial and mesenchymal markers, an immunohistochemistry analysis for E-cadherin and Vimentin was performed. This analysis confirmed the canonical expression of E-cadherin on membranes of acinar and ductal epithelial cells in healthy tissue (Figure 2A), whereas Vimentin was restricted to stromal cells and vascular endothelium (Figure 2B). Striated ducts, characterized by cuboid epithelium with eosinophilic cytoplasm, and pseudo stratified interlobular ducts maintained their structural integrity, with no evidence of EMT. In the PA cases, E-cadherin was localized at the cell membrane of epithelial cells arranged in cord-like structures. In areas undergoing EMT, its expression was reduced, and the typical membranous localization pattern was lost (Figure 2C). Vimentin was widely expressed in both myxoid stroma and epithelial subpopulations, suggesting a hybrid phenotype (Figure 2D). Finally, in the SSC, E-cadherin expression was markedly reduced, remaining detectable only in isolated residual clusters of epithelial cells (Figure 2E), whereas Vimentin expression was increased in tumor cells and, more prominently, in the stromal compartment (Figure 2F).

### 2.3. Dynamic Remodeling of the Immune-Vascular Niche in the Tumor Microenvironment

The integrated analysis of angiogenic (CD31, VEGF) and immune (MCT, CD163, and CD300a) markers revealed a differential dynamic profile between healthy PA and SCC tissues (Table 1).

CD31, an endothelial marker and indicator of microvascular density [25], showed maximum expression in AP (++++) (Figure 3C) and minimum levels in healthy tissue (+) (Figure 3A) and SCC (+) (Figure 3E). This pattern suggests that neovascularization is more orderly and efficient in the benign lesion, whereas a disorganized and often necrotic vascular environment is observed in the carcinoma. VEGF, the main pro-angiogenic factor [26], showed a surprisingly higher expression in healthy tissue (++) (Figure 3B) than in adenoma (+) (Figure 3D) and carcinoma (+) (Figure 3F). This result may reflect compensatory mechanisms or a delayed deregulation of angiogenic signaling in neoplasms.

To study the stroma-immune cell interaction, immunofluorescence analysis was performed to assess the expression of MCT and CD163 (Figure 4). The results showed that the presence of MCT is minimal in healthy tissue (+) (Figure 4B), markedly increased in pleomorphic adenoma (++++) (Figure 4D), and decreased in squamous carcinoma (++) (Figure 4F). These findings possibly suggest an active role of MCs in the benign proliferative phase and a relative loss or dysfunction in the malignant context. However, the reduced number of MCs observed in SSC may not exclusively reflect a decreased recruitment or survival, but could also indicate a complete degranulation that occurred earlier in tumor progression, thereby masking their actual presence in the tissue. This finding is consistent with previous studies attributing to tryptase a pro-angiogenic effect mediated by endothelial PAR-2 receptor activation [27]. In parallel, CD163—a marker of M2 macrophages involved in immunosuppression and tissue remodeling [28]—was detected in all three tissue types, with a moderate increase in squamous carcinoma (++) (Figure 4E), indicating a progressive activation of the stromal immune response.

### 2.4. Aberrant Co-Localization of MCs Markers in Ductal Epithelium: Immune-Epithelial Plasticity in Salivary Gland Neoplasms

In PA (Figure 5A–C), immunostaining revealed CD300a expression not only in stromal immune cells but also within the epithelial cells lining some residual excretory ducts. Here, partial co-localization of CD300a and MCT was observed in ductal epithelial cells, suggesting potential phenotypic modulation or interaction between MC-derived mediators and epithelial components in the tumor microenvironment. Additionally, CD163-positive M2 macrophages expressing CD300a were frequently observed within the PA stroma, supporting a multifaceted role of CD300a in modulating local immune responses. In SCC samples (Figure 5D–F), the glandular architecture was largely disrupted, with rare MCT-positive MCs scattered throughout the tissue and no identifiable excretory ducts. CD300a expression was markedly reduced and limited to isolated cells, with minimal co-localization with MCT. The down regulation of both markers and the loss of ductal structures reflect significant alterations in the tumor immune microenvironment and the disruption of normal tissue homeostasis.

To further explore the localization of MCT and CD300a within salivary gland tissues, double immunofluorescence staining was performed using cytokeratin 18 (CK18) in combination with MCT and CD300a. The staining showed apparent co-localization of CK18 with both MCT and CD300a in ductal epithelial cells, suggesting a possible epithelial expression of these markers (Figure 6 and Figure 7).

Notably, in SCC, despite the widespread architectural alteration and absence of organized ductal structures, the residual intralobular ducts still showed partial co-localization of CK18 with MCT and CD300a. This suggests that even in advanced malignant forms, a subset of epithelial cells may retain or abnormally acquire features typically associated with MCs, probably reflecting a form of epithelial plasticity or tumor-associated immune mimicry.

## 3. Discussion

The present study outlines a dynamic immunophenotypic landscape in salivary gland tissues, highlighting how physiological epithelial–mesenchymal compartmentalization (E-cadherin+/Vimentin+) changes during neoplastic transformation. In healthy tissues, the canonical localization of CD300a and MCT in periductal MCs reflects their established role in immune surveillance, supported by the coordinated expression of endothelial (CD31) and vascular growth factor (VEGF) markers. This balance, in which MCs act as immune sentinels and vascular architecture maintains strict homeostasis, represents the morpho-functional substrate upon which neoplastic alterations are grafted, in line with recent observations on stromal dynamics in exocrine glands [29,30].

In PA, the aberrant acquisition of MCs markers (MCT, CD300a) by ductal epithelial cells (CK18) suggests a novel phenotypic plasticity that transcends the traditional dichotomy between epithelial and immune lineages. The precise co-localization of MCT and CD300a with CK18, observed in most ducts, could result from cellular trans differentiation mechanisms, as documented in other solid tumors [31], or from the internalization of exosomes containing MCs mediators [32]. This phenomenon is associated with efficient vascular remodeling, characterized by high micro vascular density (CD31++++), which support benign tumor growth through ordered neo angiogenesis [33]. Activation of the PAR-2 receptor by MCT, known to promote endothelial proliferation [27], could mediate this vascularization, while CD300a, an inhibitory receptor unexpectedly expressed in the epithelium, could locally modulate immune activation, preventing harmful inflammatory responses [34,35]. This state of “adaptive equilibrium” between epithelial, immune and vascular components could explain the slow clinical progression of PA, despite its high cellularity, and is reflected in studies on the plasticity of salivary stem cells in reparative contexts [36].

In contrast, in SCC, the collapse of this multifunctional interface marks the transition to a chaotic and immunosuppressive microenvironment. The loss of ductal architecture, the down regulation of CD300a/MCT and the dominance of M2 macrophages (CD163++) might reflect a hypoxic environment, characterized by EMT, dysfunctional vascularization (CD31+) and altered angiogenic signaling (reduction in VEGF) [37]. The drastic reduction in VEGF in SCC compared to healthy tissue, although counterintuitive, could indicate a failure of compensatory mechanisms regulating physiological angiogenesis, which may be replaced by alternative angiogenic pathways primarily dependent on angiopoietins and their receptors (e.g., Tie2), rather than VEGF signaling [38]. However, the inability of the latter to induce effective vascular maturation results in immature and permeable vessels, typical of advanced malignancy [39]. At the same time, the accumulation of M2 macrophages in the desmoplastic stroma, associated with immunosuppression and matrix remodeling, creates a vicious circle that promotes invasion and metastasis [28], a phenomenon already described in salivary gland carcinomas with an unfavorable prognosis [40,41,42,43].

The residual persistence of CK18/MCT/CD300a co-localization in the rare intralobular ducts of SCC represents an element of continuity with the PA phenotype, suggesting the selective survival of epithelial subpopulations with hybrid identity. These cells, potentially endowed with stem or apoptosis-resistant properties [44], could act as a reservoir for tumor progression or post-therapeutic recurrence. Their coexistence with a predominantly immunosuppressive microenvironment raises questions about their ability to interact with residual immune cells, modulating anti-tumor responses or promoting evasion mechanisms, a process also hypothesized in models of tumor plasticity [45].

The observation of immune markers in non-hematopoietic epithelial compartments challenges the strict histogenetic classification of salivary gland tumors. The apparent co-expression of MCT and CD300a in the epithelium of PA might reflect either technical proximity to infiltrating MCs or a potential uptake of MCs-derived proteases such as MCT by tumor cells, rather than true endogenous expression. This phenomenon has not been directly reported as epithelial expression of MTC in the literature; on the other hand, previous studies described MCs infiltration within tumor tissues [46,47]. Such uptake or close association could allow tumor cells to mimic certain MC functions, such as modulating protease activity or inflammatory signals, thereby manipulating the microenvironment to favor tumor survival. Similarly, the downregulation of CD300a in SCCs could deprive epithelial cells of a proliferation control mechanism, contributing to malignant progression [48].

However, the expression of MCT in epithelial cells is unexpected and raises important questions. Although it is natural to question the veracity of such an unexpected finding, the experiments were repeated enough times to make the likelihood of this observation being merely an artifact very low. Nonetheless, given the novelty of the result, we cannot entirely exclude the possibility that it represents a technical artifact or a misidentification due to the close association of MCs with epithelial structures; alternatively, it may reflect a rare instance of lineage plasticity or uptake of MC-derived proteins by epithelial cells [49]. Moreover, CD300a is predominantly expressed on hematopoietic cells, including myeloid cells and certain lymphoid populations, but is not typically associated with granulocytes such as eosinophils and basophils and MCs themselves [50]. Its apparent presence in epithelial cells may indicate either ectopic expression or interactions within the tumor microenvironment that modulate immune-related molecules in non-hematopoietic cells [20]. An additional consideration is that the reduced detection of MCs in malignant tissues could also result from complete degranulation occurring earlier during tumor progression, which would obscure their presence in histological analyses. This possibility highlights the importance of complementing immunofluorescence with functional assays or temporal studies to determine whether the observed epithelial expression reflects true cellular reprogramming or secondary uptake of MC-derived products.

The therapeutic implications of these results are multiple. First, the targeting of MCT/PAR-2 in PA could offer an approach to inhibit angiogenesis without completely destabilizing the microenvironment while partially preserving tissue homeostasis. Secondly, the reactivation of CD300a in SCC through specific agonists could restore inhibitory signals to counter uncontrolled proliferation. Finally, the identification of hybrid epithelial subpopulations (CK18/MCT) would pave the way for targeted therapies to eliminate resistant residual cells, reducing the risk of recurrence.

However, several methodological limitations need to be considered. The small sample size (10 cases per group), although adequate for qualitative analysis, may limit the generalizability of results, especially in rare tumors such as SCC of the salivary glands. In addition, immunofluorescence, while allowing the visualization of co-localization, does not provide direct evidence of functional interaction or autonomous protein synthesis [51]. Complementary approaches, such as in situ RNA hybridization or single-cell transcriptomic analysis, would be necessary to confirm de novo gene expression and clarify the cellular source of CD300a.

Further studies should explore the biological significance of observed epithelial plasticity. The hypothesis that epithelial cells may acquire immune-like functions to modulate the microenvironment requires functional validation. A promising approach would involve co-culture systems of MCs and epithelial cells, which have been previously used to study intercellular signaling and paracrine modulation [52,53]. These models could be complemented by treatments with MCT or CD300a inhibitors to assess their role in epithelial phenotypic modulation. In vivo mouse models of PA and SCC could clarify whether the aberrant expression of MC-associated markers in epithelial compartments directly contributes to tumor growth, immune evasion, or drug resistance. Furthermore, the role of CD300a in epithelial cells remains elusive: although classically known to inhibit activation signals in MCs and other myeloid lineages [54], its intracellular signaling pathways in epithelial neoplasms have yet to be elucidated.

From the diagnostic point of view, these results require a critical review of current histopathological criteria. The presence of immune markers in epithelial cells may induce classification errors, especially in lesions with hybrid or atypical phenotypes. The integration of multimarker panels (MCT, CD163, CD300a, CK18) could improve diagnostic accuracy by distinguishing between benign proliferations with active stroma and low-differentiation carcinomas. From a diagnostic perspective, the present findings underscore the importance of cytological and immunophenotypic profiling in salivary gland pathology. The ability to detect immune-related markers such as MCT and CD300a within epithelial compartments provides a potentially valuable adjunct to conventional histopathology, especially in cases where morphological features alone may not suffice for accurate classification. Cytological analysis, particularly when integrated with multimarker immunofluorescence panels, could allow early recognition of hybrid or transdifferentiated epithelial populations, improving diagnostic sensitivity in borderline or ambiguous lesions. Such an approach might reduce both time and cost compared to more invasive procedures like repeated excisional biopsies or extended molecular testing, while increasing diagnostic efficiency through rapid in situ evaluation of cellular identity. Nevertheless, the extent to which cytological findings can substitute for histological gold standards remains limited: while they may reliably differentiate reactive conditions, PA, and SCC at a preliminary stage, confirmatory tissue-based assays are still required to validate lineage reprogramming phenomena and to exclude technical artifacts. In this sense, the integration of cytology with targeted immunoprofiling could serve as a cost-effective triage tool, streamlining the diagnostic workflow, reducing turnaround times, and selectively directing cases toward more advanced molecular or transcriptomic analysis when necessary.

## 4. Materials and Methods

### 4.1. Sample

The study sample consisted of 30 histological samples, including 10 normal salivary glandular tissues, 10 cases of pleomorphic adenoma and 10 cases of primary squamous cell carcinoma. All specimens were provided by the Histopathology Laboratory of the PROMISE Department, Palermo Medical School, Italy. All patient biopsies were obtained before the start of any pharmacological or surgical treatment, as they were necessary for diagnostic purposes. The cohort included both male and female patients, with a mean age of 41 years. The identity of the patients was withheld from the researchers. Since the samples were archived and dated more than ten years before the start of the study, patient consent was not required in accordance with institutional ethical guidelines.

### 4.2. Histopathology

To prepare the tissue for histological examination, the tissue sections were obtained by microtome cutting and deparaffinized by sequential immersion in xylene, followed by rehydration through a graded ethanol series. Sections were stained with hematoxylin and eosin (H&E): slides were immersed in hematoxylin for 5 min, washed in distilled water, and counterstained with eosin for 2 min. After dehydration and clearing in xylene, the slides were mounted with DPX mounting medium. Stained slides were evaluated under a light microscope (Axioscope 5/7 KMAT, Carl Zeiss, Oberrkochen, Germany) connected to a digital camera (Microscopy Camera Axiocm 208 color, Carl Zeiss) for digital images capture.

### 4.3. Immunohistochemistry

For immunohistochemical analysis, deparaffinized tissue sections were subjected to antigen retrieval using Tris-EDTA buffer (pH 9.0) (Sigma Aldrich, St. Louis, MO, USA) or 10 mM trisodium citrate (pH 6.0) (Sigma Aldrich, St. Louis, MO, USA) with 0.05% Tween-20 (Bio-Rad Laboratories, Hercules, CA, USA). Heat-induced antigen retrieval was performed in a water bath at 75 °C for 8 min and then immersed in acetone at −20 °C for 5 min to prevent sections from detaching from the slides. To block endogenous peroxidase activity and reduce non-specific staining, tissue sections were incubated with 3% hydrogen peroxide in distillated water for 10 min. After a 5 min wash with phosphate-buffered saline (PBS) (pH 7.4) (Ventana Medical Systems, Inc., Innovation Park Drive Tucson, AZ, USA), the sections were immune-stained using the Novolink Polymer Detection System (RE7140-CE, Leica Biosystems, Deer Park, CA, USA), based on the biotin–streptavidin labeling methodology.

Sections were then incubated overnight at 4 °C with primary antibody (Table 2) targeting E-caderin and Vimentin. After washing with PBS, the tissue sections were incubated with appropriate horseradish peroxidase (HRP)-conjugated secondary antibody for 30 min at room temperature. Immunoreactivity was visualized using the IHC Select^®^ HRP/DAB kit (EMD Millipore Corp., Burlington, MA, USA), following the manufacturer’s protocol. The sections were counterstained with hematoxylin, dehydrated, and mounted with a coverslip. Observations were made using a Microscope Axioscope 5/7 KMAT (Carl Zeiss, Oberrkochen, Germany), connected to a Microscopy Camera Axiocam 208 color (Carl Zeiss, Milan, Italy).

### 4.4. Immunofluorescence

Immunofluorescence analysis was performed to evaluate the localization of protein expression. Deparaffinized tissue sections underwent antigen retrieval using 10 mM trisodium citrate with 0.05% Tween-20 at 75 °C for 8 min. The sections were blocked with 3% bovine serum albumin (BSA) in PBS for 30 min at room temperature to prevent non-specific binding. Primary antibody against CD300a, MCT, CK18, CD163, CD31, and VEGF were applied at the appropriate dilutions (see Table 1) and incubated overnight with secondary antibodies anti-mouse (IgG Atto 488, 62197, Sigma) or anti-rabbit (IgG Atto 647, 40839, Sigma), diluted 1:100 on PBS) for 1 h at room temperature, protected from light. After washing with PBS (pH 7.4), nuclear staining was performed using DAPI Fluka (Sigma-Aldrich, St. Louis, MO, USA, 32670, diluted 1:1000 in PBS) and incubated for 15 min at room temperature. The slides were mounted with PBS and analyzed using a Leica Confocal Microscope TCS SP8 (Leica Microsystems, Heidelberg, Germany). High-resolution images were captured.

### 4.5. Evaluation of the Expression of Immunohistochemical Markers

The expression of angiogenic (CD31, and, VEGF) and immune (MCT, CD163 and CD300a) markers were assessed by immunofluorescence analysis on tissue sections from normal salivary glands, AP and SCC. The staining intensity was semi-quantitatively classified into four levels: weak (+), moderate (++), marked (+++), and strongly positive (++++), based on the distribution and intensity of the signal observed under the microscope. Evaluations were carried out blindly by two independent observers.

## 5. Conclusions

This study shows progressive modifications in the tissue expression of E-cadherin, Vimentin, MCT and CD300a in normal salivary glands, PA and SCC, delineating a pathobiological continuum driven by dynamic interactions between epithelial and immune-cell components. The novel co-localization of MCs markers MCT in ductal epithelial cells, particularly evident in PA, suggests phenotypic plasticity or immune mimicry, with implications for tumor microenvironment modulation. The integrated analysis of angiogenic (CD31, VEGF) and immune (CD163) markers revealed a differential vascular remodeling, characterized by neoangiogenesis ordered in the benign context and stromal disorganization in the malignancy, related to mechanisms of immunosuppression and invasive progression. The unusual expression of CD300a in the ducts of PA, associated with a preserved MCs density, emerges as a potential indicator of active epithelial regulation, with prognostic and diagnostic value in the distinction between benign and malignant lesions. These results underline the need to re-evaluate traditional histopathological criteria, integrating multi-marker profiles to identify hybrid phenotypes and resistant cell niches. Future studies on extended case studies should validate the functional role of CD300a in the homeostasis and salivary tumorigenesis, exploring its potential as a therapeutic target or biomarker of cellular plasticity. The collected evidence provides the basis for a reassessment of biological and metabolic processes of salivary glands, in which the immune-epithelial cross-talk represents a neglected driver of neoplastic progression, opening translational perspectives in differential diagnosis and targeting of the tumor microenvironment.

## Figures and Tables

**Figure 1 ijms-26-10199-f001:**
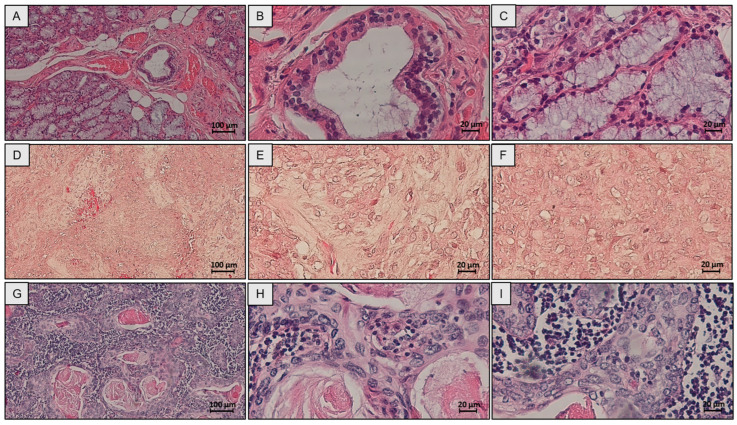
Hematoxylin-eosin staining (H&E) of salivary gland tissue. (**A**–**C**) Normal salivary glands, (**D**–**F**) pleomorphic adenoma, and (**G**–**I**) squamous cell carcinoma. Scale bar: 100 µm, magnification 100× (**A**,**D**,**G**), scale bar 20 µm, magnification 400× (**B**,**C**,**E**,**F**,**H**,**I**).

**Figure 2 ijms-26-10199-f002:**
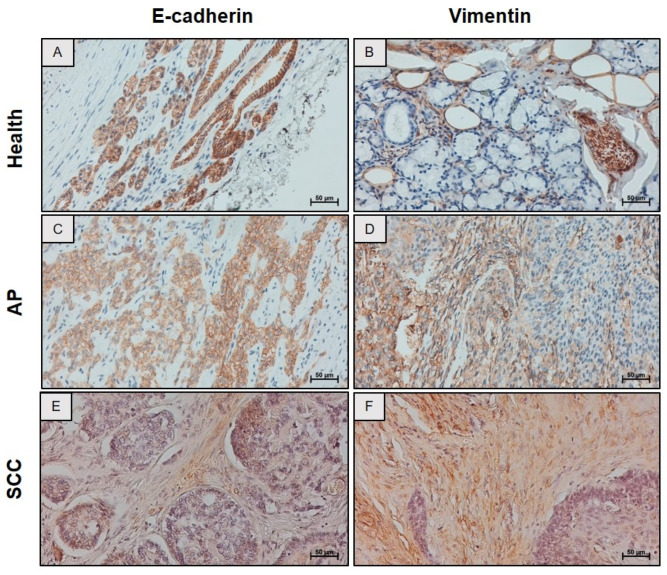
Immunohistochemistry (IHC) results for E-cadherin and Vimentin in salivary gland tissue. Representative images of: (**A**,**B**) Normal salivary glands (health) (**C**,**D**) pleomorphic adenoma (AP), and (**E**,**F**) squamous cell carcinoma (SCC). Magnification 200×. Scale bar: 50 µm.

**Figure 3 ijms-26-10199-f003:**
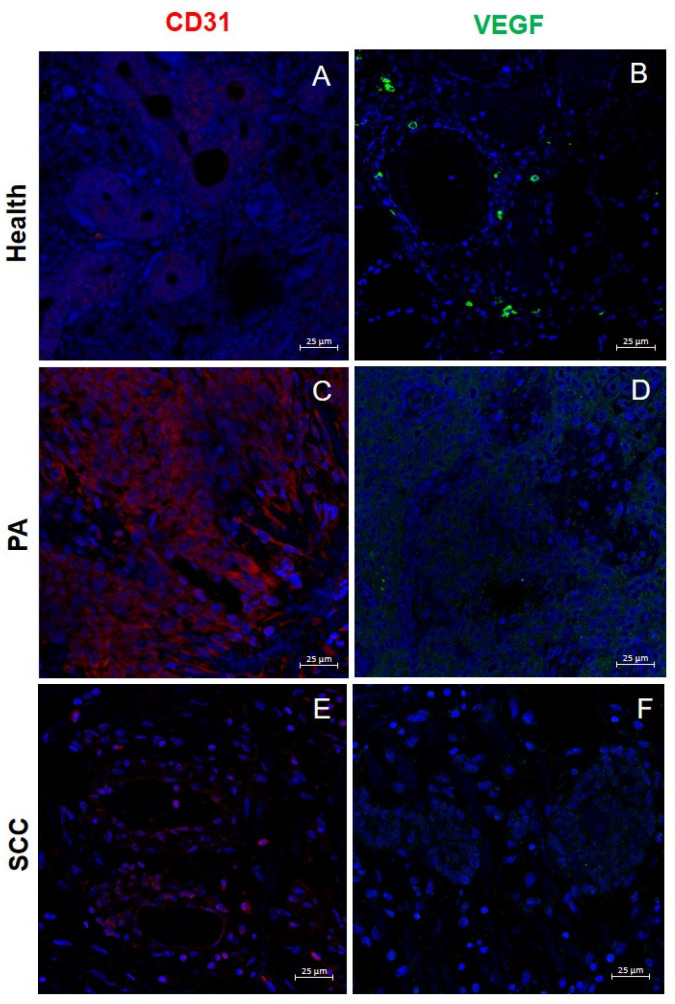
Immunofluorescence (IF) staining of CD31 (red) and VEGF (green) in salivary gland tissue. (**A**,**B**) Normal salivary glands (health), (**C**,**D**) pleomorphic adenoma (PA), and (**E**,**F**) squamous cell carcinoma (SCC). Magnification 400×. Scale bar: 25 µm.

**Figure 4 ijms-26-10199-f004:**
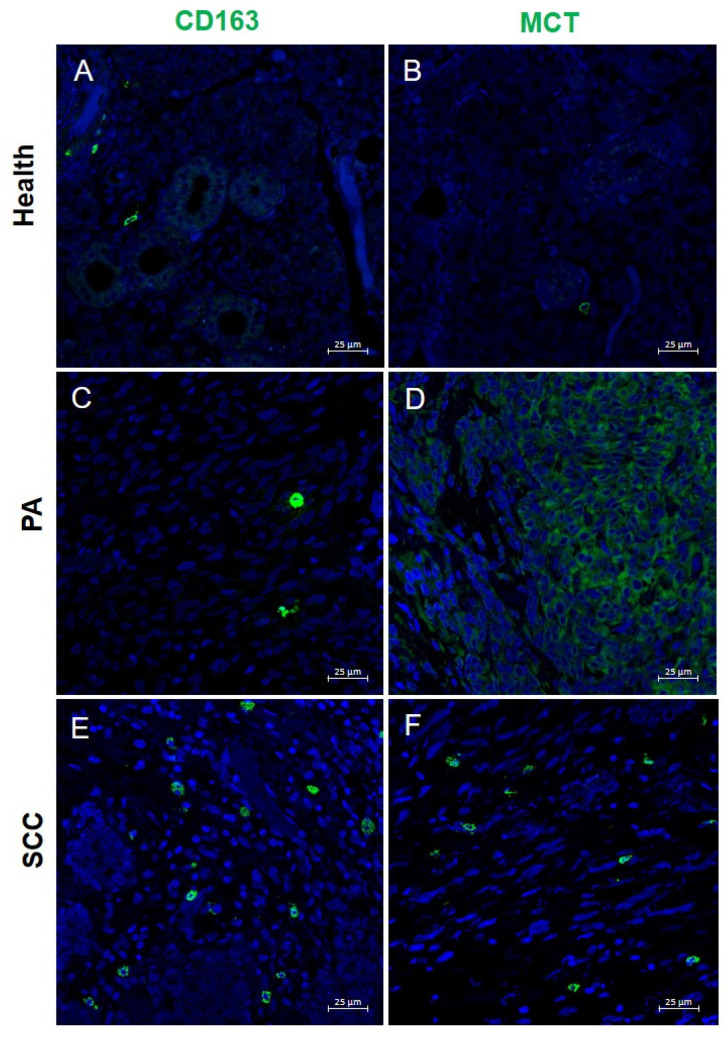
Immunofluorescence (IF) staining of CD163 and MCT (green) in salivary gland tissue. (**A**,**B**) Normal salivary glands (health), (**C**,**D**) pleomorphic adenoma (PA), and (**E**,**F**) squamous cell carcinoma (SCC). Magnification 400×. Scale bar: 25 µm.

**Figure 5 ijms-26-10199-f005:**
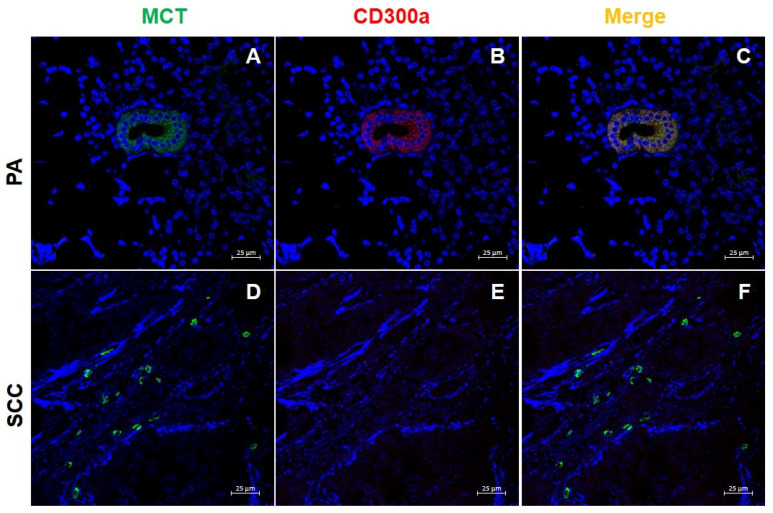
Immunofluorescence (IF) staining of MCT (green), CD300a (red), and merge (yellow) in salivary gland tissue. (**A**–**C**) pleomorphic adenoma (PA), and (**D**–**F**) squamous cell carcinoma (CSS). Magnification 400×. Scale bar: 25 µm.

**Figure 6 ijms-26-10199-f006:**
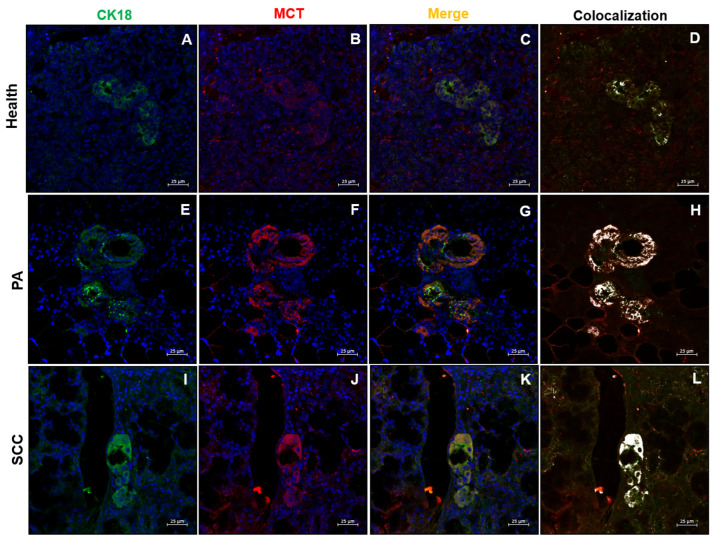
Immunofluorescence staining for CK18 (green), MCT (red), merge (yellow), and co-localization (white) in salivary gland tissue. (**A**–**D**) Normal salivary glands (health), (**E**–**H**) pleomorphic adenoma (PA), and (**I**–**L**) squamous cell carcinoma (SCC). Magnification 400×. Scale bar: 25 µm.

**Figure 7 ijms-26-10199-f007:**
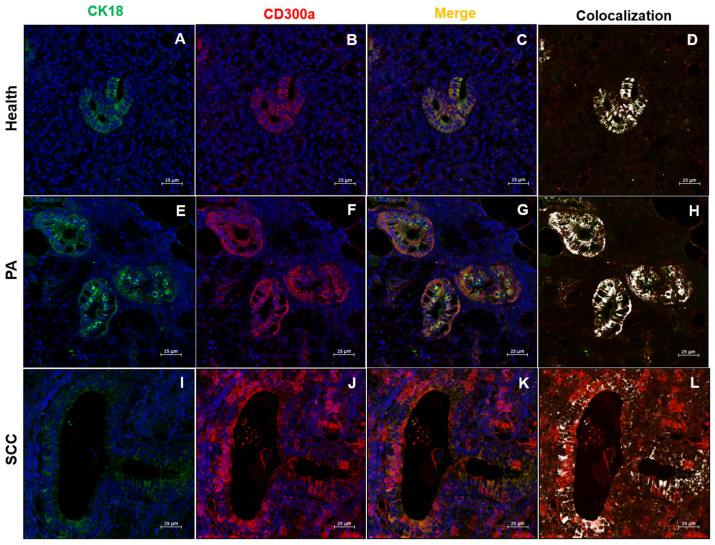
Immunofluorescence (IF) staining for CK18 (green), CD300a (red), merge (yellow), and co-localization (white) in salivary gland tissue. (**A**–**D**) Normal salivary glands (health), (**E**–**H**) pleomorphic adenoma (PA), and (**I**–**L**) squamous cell carcinoma (SCC). Magnification 400×. Scale bar: 25 µm.

**Table 1 ijms-26-10199-t001:** Comparative table of marker expression in the three tissue groups.

Marker	Healthy	AP	SCC
CD31	+	++++	+
VEGF	++	+	+
CD163	+	+	+
MCT	+	++++	++
CD300a	++	+++	++++
CD300a/MCT in the ducts	+	++++	+

AP—Pleomorphic adenoma; SCC—Squamous cell carcinoma. Legend: + weak; ++ moderate; +++ marked; ++++ strongly.

**Table 2 ijms-26-10199-t002:** List of the antibodies and dilutions used for analysis of immunohistochemistry (IHC) and immunofluorescence (IF).

Primary Antibody	Features	IHC	IF
Anti-E-Cadherin (36B5)	PA0387, mouse monoclonal, Leica Biosystems, Sheffield, UK	1:100	-
Anti-Vimentin	NCL-L-VIM-V9, mouse monoclonal, Leica Biosystems, Sheffield, UK	1:100	-
Anti-Mast Cell Tryptase	ab2378, mouse monoclonal, abcam, Cambridge, UK	-	1:100
Anti-CD300a	ab230339, rabbit polyclonal, abcam, Cambridge, UK	-	1:100
Anti-Mast Cell Tryptase	ab134931, rabbit polyclonal, abcam, Cambridge, UK	-	1:100
Anti-CK18	SAB4200855, mouse monoclonal, Sigma-Aldrich, St. Louis, MO, USA	-	1:100
Anti-CD163 (Clone 10D6)	NCL-L-CD163, mouse monoclonal, Leica Biosystems, Sheffield, UK	-	1:100
Anti-CD31	ab182981, rabbit polyclonal, abcam, Cambridge, UK	-	1:2000
Anti-VEGF	MAB2932, mouse monoclonal, Bio-Techne’s R&D Systems, Minneapolis, MN, USA	-	1:100

## Data Availability

The datasets generated during and/or analyzed during the current study are contained within the article.

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
