# Peer review of "Epithelial-Immune Cell Crosstalk in Salivary Gland Tumors: Implications for Tumor Progression and Diagnostic Assessment"

_ijms, 2025, doi:10.3390/ijms262010199_

Round 1
Reviewer 1 Report
Comments and Suggestions for Authors
This manuscript presents a comprehensive immunohistochemical and immunofluorescence analysis of epithelial-immune interactions in salivary gland tumors, with a focus on pleomorphic adenoma (PA) and squamous cell carcinoma (SCC). The study is well-structured, hypothesis-driven, and addresses a clinically relevant topic with potential diagnostic and therapeutic implications. The findings suggest a novel form of immune-epithelial plasticity, particularly the aberrant expression of mast cell markers (MCT, CD300a) in ductal epithelial cells, which challenges conventional histogenetic models.
Weaknesses and Suggestions for Improvement:
- Sample Size and Generalizability
Only 10 cases per group were analyzed. While this is understandable for a rare tumor type, the small cohort limits the robustness of conclusions, particularly regarding the novel observation of epithelial CD300a and MCT expression. The claims would be more convincing with validation in a larger series or through complementary molecular techniques (e.g., in situ hybridization, single-cell RNA-seq).
- Methodological Limitations
The authors acknowledge the possibility of technical artifacts or protein uptake rather than genuine epithelial expression. However, the manuscript currently presents these findings as though they were definitive. Stronger caution in wording is necessary throughout the results and conclusions.
- Functional Validation Missing
The proposed mechanisms (immune mimicry, phenotypic transdifferentiation) remain speculative. While functional studies may be beyond the scope of this work, the authors should more clearly delimit speculation versus demonstrated findings.
- Figures and Quantification
Many conclusions rely on semi-quantitative assessments (+ to ++++). Representative images are provided, but quantification (e.g., counting positive cells, measuring fluorescence intensity) would strengthen the analysis.
5. The introduction could be shortened slightly to avoid redundancy on EMT pathways.
Author Response
Question 1 Answer: We fully acknowledge the limitation regarding the small sample
size. This is mainly due to the intrinsic rarity of this tumor type, which makes the
collection of additional cases particularly challenging. Nevertheless, we agree that the
limited cohort affects the robustness and generalizability of our findings, especially
concerning the novel observations of epithelial CD300a and MCT expression.
Importantly, our study represents a first step, and we do not intend to stop here. We
are already planning to enlarge the cohort by including further cases as they become
available, and to strengthen our observations through complementary molecular
approaches such as in situ hybridization and other advanced techniques. These future
analyses will help to validate and expand the significance of our current results.
Furthermore, we are establishing new collaborations with international
histopathology institutes to enhance the likelihood of acquiring additional cases.
Question 2 answer: We appreciate this important observation. We agree that, for
intellectual honesty, it is essential to acknowledge the possibility of technical artifacts
or protein uptake, particularly since we are reporting a novel finding. This was
precisely the reason why, after first observing this unexpected result, we repeated
the immunofluorescence experiments multiple times, far more than our usual
practice, before including them in the manuscript. The consistency of these repeated
experiments, supported by statistical analysis, gave us confidence to report the data.
Nevertheless, we fully accept the referee’s point that the current text may present
the findings as more definitive than intended. We will revise the Results and
Conclusions sections to adopt more cautious wording, making clear that while our
data strongly support genuine epithelial expression, the possibility of artifact cannot
be entirely excluded at this stage. The paragraph was mofied.
Question 3 answer: We thank the referee for this valuable remark. We fully agree
that the proposed mechanisms, such as immune mimicry and phenotypic
transdifferentiation, remain speculative at this stage. Our intention was not to
present them as demonstrated facts but rather as possible biological interpretations
that may explain our observations. We acknowledge that functional validation would
be necessary to substantiate these hypotheses, but such experiments are beyond the
scope of the present work.
Question 4 answer: We thank the referee for this constructive suggestion. In this first
step, our primary aim was to highlight the qualitative aspect of the findings, and we
believe the immunofluorescence images included in the manuscript clearly illustrate
the aberrant positivity of CD300a and MCT in ductal epithelial cells.
That said, we fully agree that quantitative approaches such as counting positive cells
or measuring fluorescence intensity would add further strength to the analysis. We
will certainly take this valuable recommendation into account in our future work,
where we plan to expand the cohort and complement the qualitative observations
with robust quantitative assessments, such as WB.
Question 5 answer:
The introduction was reformulated and reduced as the referee’s suggestion
Reviewer 2 Report
Comments and Suggestions for Authors
Best regards, I hope you are doing well today. I am writing to congratulate you on your work and the results obtained regarding in situ molecular expression in tissue samples, as well as the great work that went into producing this article. After reviewing your work, I would like to highlight the great effort and dedication of all the participants. I will share some suggestions based on the observations I found.
In the abstract section, it is noted that the acronym VEFG is written (line 22), but its meaning is not mentioned and there are no parentheses. Please explain what it means, whether it refers to vascular endothelial growth factor (VEGF), or which molecule.
It is recommended that the sections be reordered after the introduction, suggesting the following order: materials and methods, followed by results, discussion, and finally conclusions, to comply with the order requested by the journal. The current order of the sections affects the quality of the presentation of the document, but since this is an interesting, original project with excellent results, we kindly recommend that you make the suggested changes to the order.
Given the sampling limitation of a sample size of 10 for each group, it is suggested that the study be described as preliminary results with qualitative analysis. This may not limit the generalizability of the results.
The introduction does not mention the incidence rates of salivary gland tumors; it is only mentioned for squamous cell carcinoma. Therefore, it is suggested to mention the percentage of these tumors within malignant neoplasms and the percentage within head and neck tumors. Are there any records on the etiology of these tumors? Please indicate whether they exist.
On the other hand, they mention three points in their research hypothesis:
1) MC infiltration and MCT release could correlate with the loss of E-cadherin and the acquisition of the mesenchymal phenotype.
2) CD300a could modulate the activity of MC and epithelial cells, acting as a context-dependent immune checkpoint (inhibitory in PA, protumorigenic in SCC).
3) Co-expression of MCT and CD300a in ductal cells could predict aggressiveness and metastatic potential
I cordially suggest that you replace the words "could" and "might" in your hypothesis statements with concrete statements. Statements present a hypothesis that can be tested, and ultimately, based on the results, you can accept or reject that hypothesis. The suggestion would be as follows:
1) MC infiltration and MCT release correlate with the loss of E-cadherin and acquire a mesenchymal phenotype.
2) CD300a modulates the activity of MC and epithelial cells, acting as a context-dependent immune checkpoint (inhibitory in PA, protumorigenic in SCC).
3) Co-expression of MCT and CD300a in ductal cells predicts aggressiveness and metastatic potential.
No research objective was found in your writing, so it is suggested to raise it since the hypotheses must be accompanied by objectives and I also suggest that a research question be made to better explain the objective or objectives, for example: could the cellular and molecular synergy (CD300a) between MC and MCT be directly correlated with the loss of E-cadherin and result in mesenchymal phenotypic expression in order to be predictive of aggressiveness and metastatic potential in ductal cells? Reiterate that it is an example for the work that you did but that would give greater consistency to your results or decide whether to leave the hypothesis, but the objective and the research question are necessary.
Within the results, I suggest using arrows to indicate the locations where the molecules are most highly expressed in the tissues, as there are no guiding indicators for viewing the images. For example, in Figure 1, as in all images, the tissue type is only mentioned, but there is no detailed explanation or indicators in the image that could indicate the highly organized glandular architecture, with a clear demarcation between the epithelial and mesenchymal compartments. The serous and mucous acini, with their regular lobular arrangement, are delimited by a connective stroma rich in fibroblasts and blood vessels. This is intended to reach researchers who are not experts in histopathology. It is suggested that the structure of the images with their results be further detailed and that more brightness be added, since in some, for example, Figure 1, it becomes somewhat dark. Apply this suggestion to each figure. Also within the results of the expression levels of the different markers for the immune-vascular niche of the tissue microenvironment, I recommend considering, where appropriate, the histological grade of the patient, as well as their clinical staging, since perhaps these data can help and complement the results obtained for a more solid conclusion because more efficient comparisons could be made with the intensity of expression in the different samples and, for example, could complement the analysis of whether clinical staging influences the development of compensatory mechanisms or generates a delayed deregulation of angiogenic signaling in neoplasia.
Within the materials and methods section, it is recommended to include a table with the patients' clinical-pathological data to support the results. They only mention that they used samples from patients of both sexes and a certain average age (41 years).
The discussion is well-founded compared to the literature that is appropriate for the research and related to the variables studied. However, I suggest analyzing it from a diagnostic perspective. Today, it is essential to have a cytological study to identify the type of damaged cells and propose appropriate treatment. This is why I consider this work important and relevant in the field of malignant neoplasm histopathology. Furthermore, I suggest discussing whether the results of this research could reduce costs, time, and efficiency in the diagnosis of this pathology compared to other methods, or how far the results obtained from their work could go. I hope my suggestions and observations can help generate an improbable work, and I thank you for taking them into account without further ado. For now, I say goodbye, thanking you for your kind attention and wishing you successful work and an excellent day.
Author Response
Referee 2
1) vascular endothelial growth factor (VEGF). Corrected
2) sections reordered: Introduction, Materials and methods, Results, Discussion,
and finally Conclusions. Changed
3) in the introduction added the following: The global incidence of salivary gland
tumors (SGTs) is relatively low, estimated between 0.4 and 13.5 cases per
100,000 individuals per year. Benign tumors predominate; however, malignant
salivary gland cancers (SGCs) account for approximately 0.4 to 2.6 cases per
100,000. The incidence shows geographical variability and tends to be higher
in Western countries, where rates may reach 2.5–3.0 per 100,000.
4) The following sentence added: )
a) MC infiltration and MCT release correlate with the loss of E-cadherin and
acquire a mesenchymal phenotype.
b) CD300a modulates the activity of MC and epithelial cells, acting as a context-
dependent immune checkpoint (inhibitory in PA, protumorigenic in SCC).
C) Co-expression of MCT and CD300a in ductal cells predicts aggressiveness
and metastatic potential.
5) In The discussion the following paragraph added: From a diagnostic
perspective, the present findings underscore the importance of cytological and
immunophenotypic profiling in salivary gland pathology. The ability to detect
immune-related markers such as MCT and CD300a within epithelial compartments
provides a potentially valuable adjunct to conventional histopathology, especially in
cases where morphological features alone may not suffice for accurate classification.
Cytological analysis, particularly when integrated with multimarker
immunofluorescence panels, could allow early recognition of hybrid or
transdifferentiated epithelial populations, improving diagnostic sensitivity in
borderline or ambiguous lesions. Such an approach might reduce both time and cost
compared to more invasive procedures like repeated excisional biopsies or extended
molecular testing, while increasing diagnostic efficiency through rapid in situ
evaluation of cellular identity. Nevertheless, the extent to which cytological findings
can substitute for histological gold standards remains limited: while they may reliably
differentiate reactive conditions, PA, and SCC at a preliminary stage, confirmatory
tissue-based assays are still required to validate lineage reprogramming phenomena
and to exclude technical artifacts. In this sense, the integration of cytology with
targeted immunoprofiling could serve as a cost-effective triage tool, streamlining the
diagnostic workflow, reducing turnaround times, and selectively directing cases
toward more advanced molecular or transcriptomic analysis when necessary.